# Everybody Prune Now: Structured Pruning of LLMs with Only Forward Passes

## Abstract

Structured pruning is a promising approach to create smaller, faster large language models. However, existing methods typically rely on computing the gradient via backward passes, which can inflate memory requirements and compute costs. In this work we introduce `Bonsai`, a gradient-free structured pruning method that eliminates the need for backpropagation, significantly reducing memory requirements and compute costs while achieving state-of-the-art pruning performance. `Bonsai` uses forward-pass-only perturbative pruning to enable efficient compression of large models on a broader range of hardware configurations. Unlike existing structured pruning approaches, `Bonsai` not only achieves better compression with fewer resources, but also produces models that are twice as fast as those generated by semi-structured pruning. As a concrete demonstration, we use `Bonsai` to prune 7B and 8B models to 50% sparsity on a single A6000 GPU—a task challenging for backprop-based methods in memory-constrained settings, as they require 2-3× the memory. Our results show that removing backprop as a requirement not only enables pruning larger models on constrained hardware but can also lead to state-of-the-art efficiency and performance.

## 1 Introduction

The increasing scale of large language models (LLMs) has amplified the computational and memory requirements for their deployment and adaptation, presenting a significant barrier to widespread adoption. While structured pruning has emerged as a popular approach to create smaller, faster models (Wang et al., 2019; Xia et al., 2022a), many existing methods rely on backpropagation, which significantly inflates memory and compute costs: backward passes consume $\gtrsim 2\times$ (Bridger, 2023) the memory of a forward pass, with popular stateful optimizers like AdamW (Loshchilov & Hutter, 2017) requiring $\gtrsim 3\times$ more memory. This often makes them impractical for a wide range of practitioners who operate under resource constraints, such as students, researchers, and small organizations with limited access to enterprise-grade, multi-GPU systems.

In response to this challenge, we present `Bonsai`, a structured pruning approach that operates exclusively with forward passes through the parent model. By eliminating the need for backward passes for prun-

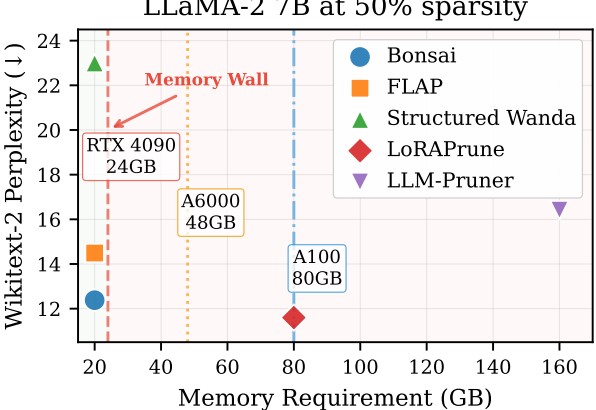

**LLaMA-2 7B at 50% sparsity**

**Figure 1: Memory requirements can make pruning methods inaccessible**. Gradient-based structured pruning methods require substantially more memory relative to forward-pass-only approaches like `Bonsai`, which can operate in memory-constrained settings (e.g., ≤24GB) while achieving superior performance among accessible methods. Results for LLaMA-2 7B at 50% sparsity on Wikitext-2.

ing, Bonsai dramatically reduces memory requirements to enable the compression of models on commodity hardware and in memory-constrained environments—potentially obviating the need for more expensive, enterprise-grade solutions. Notably, `Bonsai` maintains lower memory requirements while still outperforming existing structured and unstructured pruning approaches in terms of compression/accuracy trade-offs (Fig 1).

The key challenge in structured pruning is determining which modules (e.g., attention heads, layer dimensions) are most important for model performance and can be safely removed. Rather than computing gradients to determine module importance, `Bonsai` employs an innovative perturbative approach, estimating module importances by evaluating the performance of sub-models using only inference.

We make this perturbative approach tractable through several key innovations. First, we formulate module importance estimation as an underdetermined regression problem, allowing us to infer the importance of a large number of modules by exploring a manageable number of random sub-models. This fundamentally differs from previous perturbative approaches, which require evaluating approximately as many sub-models as there are modules (Ancona et al., 2020), making them intractable for LLMs. Second, we use informative priors derived from existing pruning approaches (Han et al., 2015; Sun et al., 2023; An et al., 2024) to guide sub-model exploration, yielding better estimates of module relevance with fewer evaluations, instead of instantiating sub-models by dropping modules with equal likelihood (Kang et al., 2023). Finally, unlike prior approaches that prune layer-by-layer (Dekhovich et al., 2021; Nova et al., 2023; Sun et al., 2023; Ling et al., 2024), `Bonsai` takes a holistic view across the entire model: modules across layers are removed and evaluated together, with relevance scores computed globally to make pruning decisions that better preserve accuracy than the sub-optimality of localized methods.

Through extensive experimentation, we demonstrate that `Bonsai`'s approach is highly efficient and effective. When compared to other structured pruning methods that also avoid backward passes, including FLAP (An et al., 2024) and a forward-only structured pruning version of Wanda (Sun et al., 2023), `Bonsai` consistently produces models with better perplexity at any given speedup target. Critically, even when compared to the state-of-the-art gradient-based structured pruning methods like LLM-Pruner and LoRAPrune, `Bonsai` produces models that majorly outperform these more memory-intensive approaches, despite using significantly less memory during pruning. Our results lead us to explore `Bonsai`'s practical utility by pruning the 3B Phi-2 (Li et al., 2023) model to a 1.8B model that performs competitively against other sub-2B parameter models on the Huggingface Open LLM leaderboard.

A key advantage of `Bonsai` is that it not only reduces memory requirements during pruning but also unlocks the opportunity for efficient subsequent post-pruning adaptation. As the pruned model is small enough to fit on the same hardware that was used for inference, practitioners can then perform parameter-efficient fine-tuning to further recover performance. This capability addresses a fundamental misconception that gradient-based methods are always superior: in many cases, `Bonsai` is a prerequisite for reaching a stage where such fine-tuning is even possible, expanding access to the entire model adaptation pipeline and yielding superior end performance. Our approach thus provides a tangible solution for practitioners who do not have access to the high-cost resources typically associated with LLM development.

## 2 Related Work

We first discuss relevant work in LLM pruning and other complementary compression methods.

### 2.1 Unstructured Pruning

While structured pruning removes entire components like layers (Xu et al., 2020; Xia et al., 2022b), dimensions of linear layers (Wang et al., 2019) or attention heads (Michel et al., 2019; Held & Yang, 2022), unstructured pruning (Han et al., 2015; Frankle & Carbin, 2018; Benbaki et al., 2023; Sun et al., 2023) removes individual parameters of the model. These approaches achieve memory savings by inducing sparsity in the model weights, but they generally do not result in tangible model speedups except when specialized hardware is available (Mishra et al., 2021). Proposed semi-structured sparsity methods (Mishra et al., 2021) such as 2:4 and 4:8 patterns can provide faster inference, but the speedup gains they achieve are far from the idealized $2\times$.

### 2.2 Gradient-Based Structured Pruning

Most existing structured pruning techniques for large (over 1B scale) language models rely on gradient computation to estimate module importance. LLM-Pruner (Ma et al., 2023) uses first-order Taylor expansion to measure the importance of coupled structures in transformers, followed by LoRA fine-tuning for performance

recovery. LoRAPrune (Zhang et al., 2024) reduces memory requirements compared to LLM-Pruner by using LoRA gradients rather than full model gradients for importance estimation, though it still requires gradient computation during the pruning process. Sheared LlaMA (Xia et al., 2024) takes a different approach by learning pruning masks during continued pre-training, integrating structured pruning with the training process. More recently, SparseLLM (Bai et al., 2024) decomposes the global pruning problem into manageable subproblems using auxiliary variables, though it maintains reliance on gradient information.

Ultimately, a fundamental limitation of gradient-based approaches is their memory requirements. Computing gradients for large models during pruning significantly increases memory usage beyond what is needed for inference alone, making these methods challenging to apply on resource-constrained hardware. In this work we show that it is possible to remove these additional memory requirements while still achieving comparable or superior performance to gradient-based pruning methods.

## 2.3 Memory-Efficient Structured Pruning Approaches

The memory constraints of gradient-based methods have motivated research into gradient-free alternatives, though such work remains more limited for LLMs. SliceGPT (Ashkboos et al., 2024) utilizes PCA to remove up to 25% of weight matrices, although it selects embedding reduction directions based on variance, which may not correlate with actual model utility for downstream tasks. In the same vein, SlimGPT (Ling et al., 2024) extends the classical Optimal Brain Surgeon (OBS) framework (Hassibi & Stork, 1992) to LLMs to reduce memory costs; however, the work admits that its layer-wise nature can lead to 'error accumulation' and an inability to use global information when pruning; we find that this makes it substantially less performant than `Bonsai`'s global approach.

For smaller language models like BERT, Nova et al. (2023) proposed Kernelized Convex Masking (KCM) for gradient-free structured pruning. Unfortunately, to prune a fully connected layer with $K$ intermediate units, KCM requires instantiating a $K \times K$ coefficient matrix for each layer. While this is feasible for BERT models, a typical LLM (AI, 2023; Touvron et al., 2023) has $K \sim 10^4 - 10^5$ which would make the size of each per-layer coefficient matrix comparable to the size of the LLM itself. In computer vision, there exist several perturbative gradient-free structured pruning techniques (Ancona et al., 2020; Dekhovich et al., 2021). However, these methods have been exclusively applied to small vision models (e.g., VGG (Simonyan & Zisserman, 2014) and WideResnet (Zagoruyko & Komodakis, 2016)) and would not scale to LLMs as-is.

Most comparably, FLAP (An et al., 2024) introduces a fluctuation-based metric that operates without gradients using statistical fluctuations in activations to estimate module importance. We show in Section 4.4 and Appendix E.3 that `Bonsai`'s flexible framework outperforms FLAP and can in fact incorporate FLAP's fluctuation-based importance metric as a special case, being a more general methodology. Finally, Probe Pruning (Le et al., 2025) introduces a different approach with "online, dynamic, structured pruning" that operates at inference time via model probing, and LoRAM (Zhang et al., 2025) trains LoRA adapters on a pruned model and uses recovered low-rank matrices with the original model for inference. Both of these methods tackle different problems to Bonsai—online pruning/fine-tuning rather than one-time permanent pruning—but they demonstrate the broader, shared trend toward memory-efficient model adaptation.

## 2.4 Positioning of `Bonsai`

Overall, the existing literature shows a clear gap: there is a need for methods that can operate within memory-constrained environments while achieving competitive structured pruning performance for LLMs. The highest-performing structured pruning approaches require computational resources that exceed what is available on single consumer GPUs.

Our approach addresses this gap by formulating structured pruning as an underdetermined regression problem that can be solved using only forward passes through the model. Key distinctions of our approach include:

- Unlike methods requiring gradient computation or auxiliary mathematical operations like Hessian storage, **`Bonsai` operates solely through model inference**.
- Rather than making layer-wise pruning decisions, **our method estimates module importance globally** across the entire model through perturbative evaluation.

**Table 1:** Landscape of resource consumption (memory and compute) of different model compression methods at training time and the inference time resource consumption of the models they deliver. ✗ represents **at least 2×** **cost** to the practitioner while ✓ → denotes a more efficient option with respect to that resource.

| Regime | Resource | Approaches | | | | |
|--------|----------|------------|-----------|--------------|-------------------|---------------|
| | | Quantization (Mixed Precision) | Distillation | Unstructured Pruning | Gradient-Based Structured Pruning | Bonsai (Ours) |
| Train | Memory | ✓ | ✓ | ✓ | ✗ | ✓ |
| | Compute | ✓ | ✗ | ✓ | ✓ | ✓ |
| Inference | Memory | ✓ | ✓ | ✓ | ✓ | ✓ |
| | Compute | ✗ | ✓ | ✗ | ✓ | ✓ |

- **Our approach requires only the memory needed for inference** plus a small overhead for storing perturbation results, making it accessible on standard hardware.

- **The regression-based formulation provides theoretical grounding for estimating module importance** from limited samples, avoiding purely heuristic approaches.

### 2.5 Additional Methods for LLM Compression

Finally, although we focus on structured pruning in this work, we note that prior research has explored various other compression schemes such as distillation (Hinton et al., 2015; Gu et al., 2023) and quantization (Xiao et al., 2023) to create smaller models from larger pre-trained ones. Similar to structured pruning, these compression methods themselves often impose significant computational burdens. For example, distillation-based techniques require using LLMs to generate large amounts of teacher data (Jiao et al., 2019; Hsieh et al., 2023). Although unstructured pruning (Frantar & Alistarh, 2023; Sun et al., 2023) and quantization have lower training-time resource demands, the models they produce either require specialized hardware to achieve speedups (Mishra et al., 2021) or may actually slow down inference due to additional computational overhead (Dettmers et al., 2022). Table 1 summarizes these characteristics across different compression approaches.

Overall, each approach has distinct hardware requirements and performance characteristics that make them suitable for different deployment scenarios. Our structured pruning approach could potentially be combined with these techniques; for instance, quantization could be applied post-pruning, and structured pruning could serve as initialization for distillation approaches. We leave such combinations for future work and focus specifically on the forward-pass-only structured pruning problem.

## 3 Methodology

We cover background in LLM pruning (§3.1), and then discuss `Bonsai`, our structured pruning method that exclusively performs inference on the parent model to prune. Figure 2 provides an overview of our approach as detailed in the following subsections: §3.2 describes our perturbative estimation; §3.3 explains our informative prior-based sampling strategy; and §3.4 discusses our iterative pruning procedure.

### 3.1 Background on Pruning, Problem Definition and Notation Setup

We assume that we are given an LLM, $\mathbf{M}_\theta$, parameterized by $\theta \in \mathbb{R}^D$. Also provided is $U$, a utility function that evaluates the model's performance on a target task. We instantiate $U$ as language modeling perplexity: Wikitext-2 training for Wikitext-2 experiments and C4 for zero-shot tasks. We are interested in pruning $\mathbf{M}_\theta$ to produce a smaller and faster but still performant (with respect to $U$) model under the constraint that we only have enough memory to run inference on $\mathbf{M}_\theta$. Even though we assume we can run $\mathbf{M}_\theta$ on available hardware, pruning can be critical for achieving latency targets, reducing compute burden, or making the model small enough to adapt to new (out-of-domain) tasks by performing gradient-based fine-tuning.

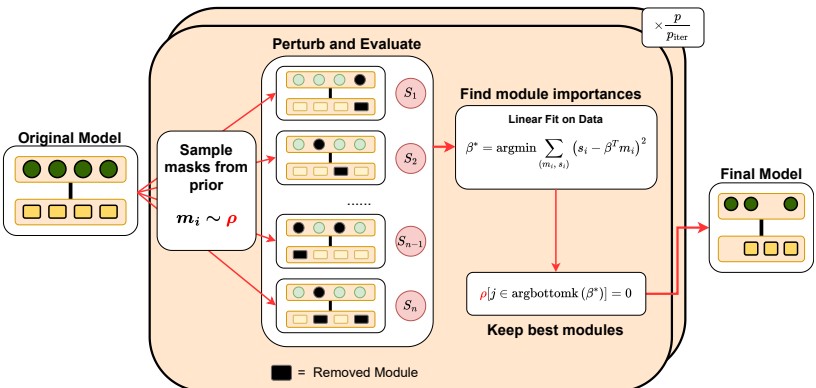

**Figure 2: `Bonsai` estimates module importance through regression on perturbative evaluations.** Rather than requiring as many sub-models as there are modules (intractable for LLMs), we solve an underdetermined regression problem: sample $n \ll N$ sub-models according to informative priors, evaluate each via for forward pass, then regress to estimate global importance scores $\boldsymbol{\beta}$. Black squares indicate removed modules.

Unstructured pruning approaches compress $\mathbf{M}_\theta$ by removing individual parameters $\theta_j$ from the model. This results in the updated model consisting of sparsified weight matrices with a smaller memory footprint. Unfortunately, the updated model may not enjoy inference speedups except when specialized hardware is available and thus can pose a compute burden during inference Mishra et al. (2021). While semi-structured variants – those that remove parameters in patterns like 2:4 or 4:8 (Mishra et al., 2021) – achieve some speedup, these are modest compared to those achieved with structured pruning.

Structured pruning takes a more modular view of the units to be removed from $\mathbf{M}_\theta$. Consider that $\mathbf{M}_\theta$ is made up of modules $\mathbf{m} = \{m_i\}_{i \in [N]}$ each with corresponding parameter counts $\mathbf{s} = \{s_i\}_{i \in [N]}$ such that $\sum_i s_i = D$. For a transformer model, $\mathbf{m}$ could consist of attention heads, dimensions of fully connected layers, or even whole layers. For simplicity, we assume that $\mathbf{m}$ is made up of non-overlapping modules. Structured pruning compresses $\mathbf{M}_\theta$ by finding accurate sub-models defined by subsets of $\mathbf{m}$: provided with $\bar{\mathbf{m}} \subseteq \mathbf{m}$, we can construct an updated model $\mathbf{M}_{|\bar{\mathbf{m}}}$ that is produced by dropping the modules not in $\bar{\mathbf{m}}$ from $\mathbf{M}$. Thus, given a sparsity target $p$, structured pruning can be cast as the following combinatorial optimization problem:

$$\mathbf{m}^* = \text{argmax}_{\bar{\mathbf{m}} \in \mathcal{F}_p} \quad U\left(\mathbf{M}_{|\bar{\mathbf{m}}}\right) \quad \text{where} \quad \mathcal{F}_p = \left\{ \bar{\mathbf{m}} \subseteq \mathbf{m} \mid \left( \sum_{[j : m_j \in \bar{\mathbf{m}}]} s_j \right) \leq (1 - p)D \right\} \tag{1}$$

$\mathcal{F}_p$ consists of all sub-models that meet the sparsity threshold. Note that, in general, not only does $\mathbf{M}_{|\mathbf{m}^*}$ have a smaller memory footprint than $\mathbf{M}$, it is also faster to run inference on it since it has fewer modules. Many structured pruning methods attempt to solve Equation 1 by gradient-guided optimization (or search) over the space of sub-models. In this work, we instead explore approaches suitable for memory-constrained settings, where computing gradients may be expensive or infeasible.

### 3.2 Estimating module relevance with only forward passes

We specifically aim to develop an approach for pruning that only relies on computing the forward pass over the model. Thus, we must solve Eq 1 using evaluations of $U$ rather than gradient-based optimization. A brute-force search over $\mathcal{F}_p$ is infeasible, as its size grows combinatorially with the model. For instance, pruning a single FC sub-layer in an LLM to 50% sparsity requires evaluating $\approx \binom{10^4}{10^3}$ subsets.

We instead propose a computationally tractable approach where we first perform a small number, $n$, of evaluations, where $n \ll |\mathcal{F}_p|$, to gather data for estimating the relevance of each module in $\boldsymbol{M}$ with respect to the metric $U$. Upon carrying out this estimation, we can greedily choose the member of $\mathcal{F}_p$ that has the highest total module relevance. Specifically, let us assume that we have estimated $\boldsymbol{\beta} = \{\beta_i\}_{i \in [N]}$ to be the relevance of each of the $N$ modules. We can generate an approximate solution to Equation 1 as:

$$\boldsymbol{m}^* \approx \boldsymbol{m}^{\text{approx}} = \text{argmax}_{\bar{\boldsymbol{m}} \in \mathcal{F}_p} \sum_{j \in \bar{\boldsymbol{m}}} \beta_j \tag{2}$$

---

**Algorithm 1** `Bonsai` Pruning Method

---

1: **Input:**
2: Model [$\mathbf{M}_\theta$], sub-models per iteration [$n_{\text{iter}}$]
3: Sparsity per iteration [$p_{\text{iter}}$], Target sparsity [$p$]
4: Module list [$\mathbf{m}$]
5:
6: **for** $l = 1$ **to** $\lceil \frac{p}{p_{\text{iter}}} \rceil$ **do**
7: $\quad \rho^l \leftarrow$ Calculate unstructured pruning metric for all modules in $\mathbf{m}$
8: $\quad \bar{\rho}^l \leftarrow$ Fix the top $(1 - 2p_{\text{iter}})$ of $\rho^l$ to $\infty$
9: $\quad$ Sample $\{\bar{\mathbf{m}}_i\}_{[n_{\text{iter}}]}$ sub-models according to $\bar{\rho}^l$
10: $\quad$ Run forward pass on each sub-model and compute $U$. Construct $\mathbb{D}^l = \{\bar{\mathbf{m}}_i, U_i\}_{[n_{\text{iter}}]}$
11: $\quad \beta^l \leftarrow$ Regress $(\mathbb{D}^l)$
12: $\quad \{m_{\text{pruned}}\} \leftarrow$ sort $\beta^l$ and drop the bottom $k$ modules that make up $p_{\text{iter}}$ fraction of the model.
13: $\quad \mathbf{m} \leftarrow$ update module list to exclude $\{m_{\text{pruned}}\}$
14: **end for**
15: **Output:** Pruned model $\mathbf{M}_{|\mathbf{m}}$

---

Eq. 2 is easily solved by sorting $\beta_j$s and greedily selecting top modules until the constraint is met. This may slightly exceed the sparsity limit, but the overshoot is negligible since $s_i \ll (1 - p)D \ \forall \ i$ in our settings.

**Estimating $\beta$:** To estimate of the module relevance scores $\boldsymbol{\beta} \in \mathbb{R}^N$, we generate and evaluate $n \ll |\mathcal{F}_p|$ sub-models, constructing a dataset $\mathbb{D} = \{\bar{m}k, Uk\} \ k \in [n]$ where $U_k = U(\mathbf{M}_{|\bar{\boldsymbol{m}}_k})$. We then frame the estimation of $\boldsymbol{\beta}$ as an under-specified regression problem:

$$\hat{\boldsymbol{\beta}} = \text{argmin}_{\boldsymbol{\beta} \in \mathbb{R}^N} \quad \frac{1}{n} \sum_{(\bar{\boldsymbol{m}}_k, U_k) \in \mathbb{D}} \left( U_k - \beta^T \alpha_{\bar{\boldsymbol{m}}_k} \right)^2 + \gamma \|\boldsymbol{\beta}\| \tag{3}$$

where $(\alpha_{\bar{\boldsymbol{m}}_k})_i = \mathbf{1}[i \in \bar{\boldsymbol{m}}_k]$, is the binary vector that has 0 at indices with modules dropped. Implementing a sub-model $\bar{\boldsymbol{m}}_k$ as a binary mask $\alpha_{\boldsymbol{m}_k}$ is key to practically realizing our approach. We never actually instantiate sub-models as this would be prohibitively expensive. Instead, we create them *virtually* by zeroing out the outputs of the parts to be pruned so they have no effect on the model output.

### 3.3 Selecting sub-models for evaluation

An important design choice is selecting the $n$ candidate sub-models for evaluation. A naive approach such as uniform sampling would be suboptimal. Take $m_i$ as a module that is critical for good performance under evaluation with $U$. Since $n < N$, it means that some modules may never be "turned on" in the list of $n$ chosen sub-models. If $m_i$ happens to be one of these masked-out modules, the resulting estimate of $\hat{\beta}_i = 0$ would in turn result in $\boldsymbol{m}^{\text{approx}}$ being a poor estimate for $\boldsymbol{m}^*$. Thus, a more informed selection is necessary for accurate and useful $\boldsymbol{\beta}$ estimates.

Given a module $m_i$, we instead set the likelihood of it being present in any of the $n$ sampled sub-models proportional to a prior $\rho_i$ which reflects its usefulness. To define $\rho_i$, we can turn to metrics from the pruning literature. Specifically, we set $\rho_i$ to be a module-level analogue of any of the pruning metrics like Wanda (Sun et al., 2023) or activation magnitude. For example, in one-hidden-layer network of dimension $d$, if activation magnitude is the prior, we compute the activation vector over multiple samples. The probability of retaining the $i$th column of $W \in R^{d \times d}$ is: $\rho_i \propto \hat{\boldsymbol{a}}_i = \frac{1}{B} \sum_b \left| \sigma\left( (W^T[i, :]) x_b \right) \right|$ where $\sigma$ is the nonlinearity. Sampling sub-models based on $\boldsymbol{\rho}$ biases selection toward high-performing models. Since $\rho_i$ can be computed via forward passes through the unmodified model $\boldsymbol{M}_\theta$, this approach remains memory-efficient. Appendix E details the priors we explored.

To enhance efficiency, we prune only the bottom $2p$ fraction of modules per layer, ranked by prior $\rho$, while keeping the top $1 - 2p$ fraction fixed.[1] This reduces the search space for sub-model evaluation. For the bottom $2p$ fraction, whenever we generate a mask $\alpha_{\boldsymbol{m}_k}$ with sparsity $p$, we also generate its complement $\alpha^c_{\boldsymbol{m}_k}$,

---

[1]$2p$ is arbitrary; practitioners can tune it for optimal performance.

**Table 2:** Reported memory consumption of different methods. The minimum amount of memory required to run a LlaMA-7B model at half precision (FP16) is 14GB. Running a forward pass with batch size of 1 using the default model sequence length of 4096 uses around 20GB of memory.

| Base Model | Forward Only | | Gradient-Based | | | |
|:---:|:---:|:---:|:---:|:---:|:---:|:---:|
| | Bonsai (Min.) | Bonsai (Faster) | LoRA Prune | Compresso | LLM-Pruner | Sheared LlaMA |
| LlaMA-2-7B | Forward + Bsz=1 | | (Zhang et al., 2024) | (Guo et al., 2023) | (Ma et al., 2023) | (Xia et al., 2024) |
| **14GB** | **20GB** | **48GB** | **80GB** | **128GB** | **160GB** | **640GB** |
| FP16 only | A6000 | 1×A6000 | 1×A100 | 4×V100 | 2×A100 | 8×A100 |

obtained by flipping the values in $\alpha_{\boldsymbol{m}_k}$ (excluding the fixed $1-2p$ fraction). Covert & Lee (2020) show that this technique helps lower variance in regression with binary inputs.

### 3.4 Iterated Pruning

Previous works on gradient-based pruning (Anwar et al., 2017; Frankle & Carbin, 2018) have shown that taking an iterated approach to pruning yields improved results over pruning directly to the target sparsity $p$. Similarly, we define an updated pruning fraction $p_{\text{iter}} < p$ and perform iter $= \lceil \frac{p}{p_{\text{iter}}} \rceil$ steps where we explore $n_{\text{iter}} = \lceil \frac{n}{\text{iter}} \rceil$ sub-models at a time. At the beginning of each iteration, we re-estimate the priors $\boldsymbol{\rho}$ for the unpruned modules and use them to select the $n_{\text{iter}}$ sub-models to evaluate.

We combine the methods from Sections 3.2, 3.3, and 3.4 to develop `Bonsai`, a gradient-free structural pruning algorithm (Figure 2). Algorithm 1 provides the full details. A note about Line 5 in our algorithm: depending on the task, we find that sampling $\bar{\mathbf{m}}_i$ and its complement $\bar{\mathbf{m}}_i^c$ helps reduce the variance of our regression estimate and leads to better results.

## 4 Experimental Details and Main Results

In this section, we empirically evaluate `Bonsai` across multiple structured pruning settings. We compare `Bonsai` + PPA against both **semi-structured pruning** (Section 4.1) and **gradient-based pruning** (Section 4.2). Next, we demonstrate that `Bonsai` can yield **compact models with strong zero-shot abilities** (Section 4.3). Finally, we compare `Bonsai` to other **forward-pass-only pruning methods** (Section 4.4) to establish competitiveness under memory constraints.

In all `Bonsai` experiments, we prune (1) self-attention heads and (2) fully connected layer dimensions, focusing on LLMs around 7B parameters. Since our goal is to aid memory-constrained practitioners, Table 2 compares memory requirements for pruning LlaMA-2-7B across methods. As can be seen, for models of our size range of interest, we can run `Bonsai` on any device with $\approx 20$GB of memory if we restrict our batch size to 1.

To reduce variance in score estimates, we average over 32 data points. A practitioner with 20GB of memory would run 32 forward passes with batch size 1, but this extends experiment runtime. Instead, we use a 48GB A6000 GPU (although not required by `Bonsai`), enabling batch sizes of 4–8 for faster experimentation.

**Introducing Post-Pruning Adaptation (PPA).** A central strength of `Bonsai` is that, by reducing models to a size that fits on commodity inference hardware depending on the sparsity level $p$ achieved, it unlocks the ability to further fine-tune (full fine-tuning or a parameter-efficient fine-tuning method like LoRA (Hu et al., 2021)) the pruned model on the same hardware. PPA is not required for `Bonsai` to be effective; but it shows that `Bonsai` does more than prune, since it expands access to to the full model adaptation pipeline in settings where gradient-based pruning is infeasible from the start.

Like many past works (Sanh et al., 2020; Xia et al., 2022b), we can combine pruning with distillation by incorporating a distillation loss in the training objective during fine-tuning of the pruned model. Let $\mathcal{L}_{\text{task}}$ be the loss function over the task data and $\mathcal{L}_{\text{distill}}$ be the distillation objective. We optimize the following post-pruning objective: $\mathcal{L}^{\text{post-prune}} = \mathcal{L}_{\text{task}} + \lambda \mathcal{L}_{\text{distill}}$. Using $i$ to index the task data, we have:
$\mathcal{L}_{\text{distill}} = \sum_i D_{\text{KL}} \left( \text{logits}^i \left( \mathbf{M}_{|\mathbf{m}^{\text{approx}}} \right) \parallel \text{logits}^i \left( \mathbf{M} \right) \right)$, where $\lambda$ is a scalar weighting that can be cross-validated. Note, distillation can be performed without significant memory overhead by *a priori* caching the logits from

the parent model **M** instead of hosting the model in memory during fine-tuning. In the subsections below, we will apply PPA **after** pruning the parent model **if the child model is small enough to allow for this**.

### 4.1 `Bonsai` **is competitive with semi-structured pruning methods**

We compare `Bonsai` to the semi-structured variant of Wanda (Sun et al., 2023). In general, structured pruning under-perform semi-structured pruning, but compensate for this in speedup.

Before fine-tuning, the Wanda 2:4 model is more accurate but slower ($1.14\times$ vs $1.58\times$) than the model from `Bonsai`. Since the `Bonsai` child model is small enough, we can perform PPA on it, resulting in improved accuracy (8.89ppl) with unchanged speedup.

Fine-tuning the semi-sparse Wanda 2:4 model is unfortunately less straightforward. It would require similar memory resources to finetune the parent model[2]; however, our setting does not have enough memory for this. We therefore have to use a parameter-efficient fine-tuning method like LoRA (Hu et al., 2021) instead. While the performance gap can be bridged by LoRA fine-tuning ($10.52 \rightarrow 8.34$), the

**Table 3: Pruning can match highly optimized small, pretrained models while delivering better speedups.** Bonsai-pruned LLaMA-2 achieves accuracy comparable to Phi-2 (a carefully engineered 3B model) with superior inference efficiency. Semi-structured pruning loses speedup benefits due to incompatible sparsity patterns.

| Model | ~Size | Fine-tune | PPL | Speedup |
|-------|-------|-----------|-----|---------|
| LlaMA-2 | 7B | ✗ | 5.11 | $1\times$ |
| Phi-2 | 3B | ✓ | 8.69 | $1.24\times$ |
| Wanda 2:4 | 3B | ✗ | 10.52 | $1.14\times$ |
| + PPA | | ✓ | 8.34 | $0.75\times$ |
| `Bonsai` | 3B | ✗ | 19.47 | $\mathbf{1.58\times}$ |
| + PPA | | ✓ | 8.89 | $\mathbf{1.58\times}$ |

adapted semi-structured model experiences a drastic slowdown ($0.75\times$), since the learned low-rank matrices cannot be merged with the original sparsified ones without reverting back to dense computation. Thus, LoRA fine-tuned Wanda 2:4 is twice as slow ($\sim 0.48\times$) than the model from `Bonsai` and similarly accurate.

In a memory-constrained setting, practitioners could opt for a pre-existing model of the target size instead of pruning a larger model. We compare the Bonsai-pruned model to Phi-2 (Li et al., 2023), a strong representative pre-existing model of similar size. **As can be seen in Table 3, Bonsai is able to generate a model that is as accurate (0.2 difference in ppl) yet significantly faster ($1.58\times$ vs. $1.24\times$ speedup), thus making it a competitive option to consider even if a model already exists at the target size.**

### 4.2 `Bonsai` **is competitive with gradient-based structured pruning**

**Table 4:** LlaMA-1 (50% sparsity) after post-pruning adaptation. [†] are results as reported by Zhang et al. (2024). All methods use PPA on proxy data (20-50K samples); `Bonsai` ahieves this with 4-8$\times$ less memory during pruning.

| Method | Wikitext-2 ↓ | BoolQ | HellaSwag | WinoGrande | ARC-e | ARC-c | Average ↑ |
|--------|--------------|-------|-----------|------------|-------|-------|-----------|
| LlaMA1-7B (Touvron et al., 2023) | 5.68 | 75.05 | 56.92 | 69.93 | 75.34 | 41.89 | 63.83 |
| LLM-Pruner[†] (Ma et al., 2023) | 16.41 | 60.28 | 47.06 | 53.43 | 45.96 | 29.18 | 47.18 |
| LoRAPrune[†] (Zhang et al., 2024) | 11.60 | 61.88 | **47.86** | 55.01 | 45.13 | **31.62** | 48.30 |
| `Bonsai` + PPA | **10.92** | **67.22** | 43.09 | **61.64** | **54.92** | 26.28 | **50.63** |

Next we compare `Bonsai` to the following gradient-based structured pruning approaches: LLM-Pruner (Ma et al., 2023) and LoRA-Prune Zhang et al. (2024). We use the reported results from Zhang et al. (2024) since none of these methods are runnable in our memory-constrained setting (Table 2). We choose to compare to these over Sheared LlaMA (Xia et al., 2024) since they have much lower memory requirements (Table 2). We prune the LlaMA-1 7B model (Touvron et al., 2023) to 50% sparsity since these approaches report their results for the LlaMA-1 model only. We compare these methods on Wikitext-2 and also on six tasks from the Eleuther

---

[2]though the child tensors are sparse, the resulting gradients and cached intermediate tensors can be dense and have the same dimensions as those of the parent model (say $M \times M$). Since `Bonsai` does structured pruning, the actual tensor sizes are shrunk (say $N \times N \mid N < M$) which reduces memory during backward passes.

**Table 5:** Phi-2 pruned to 35% sparsity and fine-tune the pruned model on small amount of the C4. We achieve strong performance compared to Phi-1.5 (trained from scratch). Since Sheared LlaMA has values absent, its MC Average would be misleading and we refrain from adding it.

| Model | Size | Generation | Multiple Choice (MC) | | | | | |
|---|---|---|---|---|---|---|---|---|
| | | GSM8k (5-shot) | ARC-c (25-shot) | Winogrande (5-shot) | Hellaswag (10-shot) | Truthful-QA (0-shot) | MMLU (5-shot) | MC Average ↑ |
| Phi-2 (Li et al., 2023) | 2.7B | 54.81 | 61.09 | 74.35 | 75.11 | 44.47 | 58.11 | 62.63 |
| Phi-1.5 (Li et al., 2023) | 1.5B | 12.43 | 52.9 | 72.22 | 63.79 | 40.89 | 43.89 | 54.74 |
| Sheared LlaMA (Xia et al., 2024) | 1.3B | Not Reported | 33.5 | 57.9 | 60.7 | Not Reported | 25.7 | * |
| Bonsai (w PPA) | 1.8B | 6.37 | 47.44 | 68.35 | 65.09 | 42.20 | 40.53 | 52.72 |
| + Reasoning Tuning | 1.8B | 27.67 | 45.56 | 68.82 | 64.51 | 42.58 | 40.97 | 52.49 |

LLM Evaluation Harness benchmark (Gao et al., 2023). The pruning signal used for the Wikitext-2 task is the same as the above experiments. For the Eleuther Harness tasks, we use language modeling performance on the C4 (Raffel et al., 2020) dataset as pruning signal. We also perform parameter-efficient fine-tuning on our pruned model using 30K 512-length sequences from this corpus. `Bonsai` and LoRAPrune use similar amounts of the C4 dataset for Table 4 (30K vs 20K samples, respectively) whilst LLM-Pruner is trained on instruction tuned data with nearly twice the amount of unique samples (50K). Find more details in Appendix A.3.

**As seen in Table 4, `Bonsai` outperforms gradient-based methods even though it exclusively uses forward passes in the pruning stage.** We attribute the superior performance of `Bonsai` to the fact that its pruning decisions are informed by directly exploring the space of sub-models whilst the other approaches resort to inaccurate proxies of module relevance in order to reduce the memory overhead of a fully gradient-based optimization approach (though not by enough to be runnable in our setting).

### 4.3 `Bonsai` can produce compressed models with strong zero-shot abilities

Considerable amounts of compute and data, beyond what is feasible for many practitioners, are needed to train LLMs with strong zero-shot capabilities (AI, 2023; Gemini Team et al., 2023). In this section, we demonstrate that `Bonsai` can empower everyday practitioners to produce strong and compact models with competitive zero-shot abilities by simply pruning bigger models on their available hardware.

We use `Bonsai` to prune a ≈3B Phi-2 model to ≈1.8B (35% sparsity). `Bonsai` hyper-parameters in this experiment are in Appendix A.4. Since it is small, the 1.8B pruned model can be fully fine-tuned on 1 A6000 GPU over 100k sequences of 2,048 tokens from the C4 dataset. As can be seen from Table 5, **our pruned model achieves strong zero-shot performance on the Hugging Face OpenLLM leaderboard (Gao et al., 2023) compared to Phi-1.5, a smaller version in the Phi series of models that was trained from scratch.**

Interestingly, one exception to the general trend of Bonsai's strong performance is GSM8K, a mathematical reasoning dataset that requires generation of long reasoning chains. In our experiments, GSM8K performance is initially lower, consistent with task-agnostic pruning behavior in prior work (Xia et al., 2024; Reda et al., 2025); pruning based on language modeling can deprioritize modules needed for specialized reasoning. We attempt to remedy the drop in reasoning ability by adding 8K GSM8K samples during PPA (resulting in a total of 108K fine-tuning samples). This boosts our model's performance on the GSM8K with almost no degradation of performance on the other tasks.

### 4.4 Bonsai is competitive with other forward pass-only, structured pruning methods

We focus our last set of experiments on comparing structured pruning methods that can be run without gradient-based optimization. We prune the LlaMA-2 7B model (Touvron et al., 2023) to 50% sparsity and evalu-

**Table 6:** Perplexity at 50% sparsity of LlaMA-{1,2}-7B on Wikitext-2 and C4.

| Dataset | Method | Sparsity | LlaMA-1/seqlen=1024 | LlaMA-2/seqlen=1024 |
|---|---|---|---|---|
| Wikitext-2 | Base Model | 0% | 5.68 | 5.11 |
| | FLAP | 50% | 17.27 | 14.49 |
| | Bonsai | 50% | 15.72 | 12.38 |
| C4 | Base Model | 0% | 7.34 | 7.04 |
| | FLAP | 50% | 24.98 | 24.0 |
| | Bonsai | 50% | 22.31 | 20.7 |

ate on the Wikitext-2 (Merity et al., 2016) validation dataset. Our module importance signal for pruning, $U$, is the language modeling performance on the training set. When measuring speedups, we consider *end-to-end latency* of running inference on `model.sequence_length` chunks of the Wikitext-2 validation set. See Table 10 (Appendix A.2) for details about the hyper-parameters used for these experiments.

Figure 1 shows that at 50% sparsity, `Bonsai` achieves the lowest perplexity among methods accessible in memory-constrained settings ($\leq$ 48GB). `Bonsai` expectedly outperforms the structured variant of Wanda (Sun et al., 2023), producing models with much lower perplexity at any fixed desired speedup over the parent model. A more competitive baseline is FLAP (An et al., 2024), which can be seen as a special case of our more general framework; here (as shown in Table 6), `Bonsai` outperforms FLAP while maintaining comparable speedups.

A key advantage of `Bonsai` is its flexibility—it can incorporate various pruning methods as informative priors. In practice, we explored three classes of priors: activation magnitude (measuring the average size of activations per module), Wanda (a weight-activation product metric shown to be effective in unstructured pruing), and fluctuation-based metrics (as in FLAP, which track statistical variation across activations. These priors provide a a quick, forward-pass-only signal of which modules are likely to be important, and `Bonsai`'s regression step then refines them into more accurate importance estimates. See Appendix E for full details on how we estimate priors. An important outcome of this is that `Bonsai` offers practitioners flexible runtime-quality tradeoffs. While the reported results use a configuration that runs in $\leq 4 hours$ (which is considerably more than FLAP's $\approx 1$ hour runtime), practitioners can configure `Bonsai` to produce structurally pruned models in as little as 15 minutes by reducing the number of perturbations, sample size, or pruning iterations, potentially at some cost to performance. Conversely, allocating more compute time generally yields better pruning outcomes. This flexibility allows practitioners to tailor Bonsai to their specific constraints, unlike methods with fixed computational profiles.

## 5 Analysis

Here we conduct ablative experiments to understand the impact of the ingredients from Section 3.

**Do we need both perturbative and regressive components of `Bonsai`?**

Figure 3 shows that both components are key to obtaining a good pruned model. Removing the estimation of module importances via regression leads to a degradation in performance (61.6 ppl $\rightarrow$ 146.6 ppl). Further degradation (146.6 ppl $\rightarrow$ 405.7 ppl) is encountered if we do not perform perturbative evaluations on the parent model but simply prune according to the prior $\rho$ as computed from the unperturbed parent model.

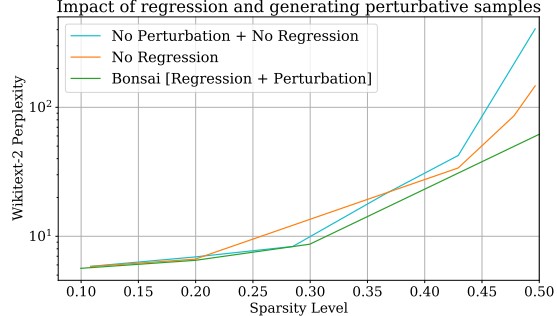

**Figure 3:** LLaMA-2 7B @ 50% sparsity (No PPA). Perturbation and regression components are both needed to make `Bonsai` effective. Experiment details in Appendix B.

**What is a reasonable number of perturbative samples?** We investigate the number of perturbative samples required to obtain good regression estimates of module importance based on Equation 3. Our results are shown in Table 7. As expected, performance improves as we increase the number of submodels explored. We note that the number of samples being explored, ns, is significantly less than the number of candidate modules at each iteration ($N \approx 70k$). Nevertheless, `Bonsai` is able to deliver a performant pruned model because of the recipes developed in Section 3.

| | ns = 50 | ns = 200 | ns = 1000 |
|---|---|---|---|
| PPL ($\downarrow$) | NaN | 61.63 | 22.09 |

**Table 7:** Wikitext-2 perplexity of LLaMA-2 7B @ 50% sparsity (No PPA). We vary the number of perturbative evaluations. Details in Appendix F.

**How much performance is recovered by post-pruning adaptation?** During iterative pruning, `Bonsai` damages the parent model by removing modules but does not perform intermittent retraining to recover lost performance since even intermediate models may be too large for fine-tuning. Even

so, as Table 8 shows, the final model produced by `Bonsai` has reasonable performance *without fine-tuning.* We attribute this to the redundancy of modules with respect to the target downstream tasks and `Bonsai`'s ability to identify good candidates for pruning. If the pruned model is small enough in size, we may perform post pruning adaptation to recover more performance, as can be seen from Table 8.

**Table 8:** Impact of PPA on LLaMA-2 7B @ 50% sparsity. Details in Appendix D.

| Method | Wikitext-2 PPL |
|---:|:---:|
| No Post-Pruning Adaptation | 19.47 |
| Post-Pruning Finetuning | 10.39 |
| + Distillation | 8.89 |

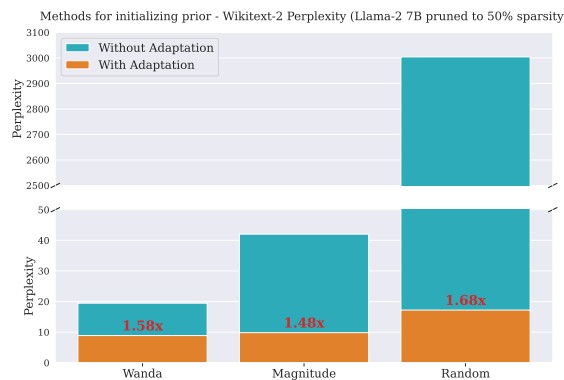

**Figure 4:** LLaMA-2 7B pruned to 50% sparsity. See Appendix E for experiment details and definitions of Wanda and Activation Magnitude priors.

**What is the impact of the choice of metric for the prior $\rho$?** We investigate three different choices of metrics for defining $\rho$. Figure 4 shows that using the module-level analogue of Wanda Sun et al. (2023) yields the best performance, both before and after post-pruning adaptation. This indicates that Wanda is a strong signal for efficiently estimating the importance of model units.

**Memory and Runtime Considerations.** `Bonsai` requires $\approx 20$GB memory versus 80-160GB for gradient-based methods (Table 2). Runtime is $\approx 4$ hours for optimal quality or $\approx 15$ minutes with reduced perturbations. Pre-adaptation results appear in Table 6 and 8; note that gradient-based baselines in Table 4 also use adaptations (Section 4.2).

## 6 Conclusion, Limitations, and Future Work

In this work, we have presented `Bonsai`, a gradient-free structured pruning method that enables efficient compression of LLMs using only forward passes. By eliminating the need for backward passes, `Bonsai` reduces memory requirements by over $2\times$, allowing practitioners to prune and adapt models on hardware configurations where gradient-based approaches would be infeasible. Through extensive experiments, we have demonstrated that `Bonsai` produces models that are small, fast, and accurate—outperforming even most of these gradient-based methods while using substantially fewer resources.

A primary limitation of Bonsai is its computational runtime. As shown in Section 5, performance improves with more sub-model exploration, slower pruning rates, and larger data samples, but these improvements come at the cost of increased runtime. While competitors like FLAP complete pruning in $\approx 1$ hour, `Bonsai`'s optimal configuration requires $\leq 4$ hours. However, Bonsai offers practitioners the flexibility to configure faster pruning schedules ($\approx$15 minutes) when speed is prioritized over achieving optimal compression quality. This runtime-quality tradeoff is a design feature that allows adaptation to various computational constraints. Additionally, since the models produced by Bonsai can guarantee optimal performance and speedups, this one-time investment becomes worthwhile when amortized over large-scale deployment.

`Bonsai` presents several promising avenues for future research. First, while our current approach samples sub-models from an informative prior, the sampling process is not completely adaptive. `Bonsai` could be further enhanced by dynamically exploring the space of sub-models based on previous evaluations. Additionally, due to our focus on forward-pass-only operations, `Bonsai` does not fine-tune the model during iterative pruning. Integration with gradient-free approaches like MeZO (Malladi et al., 2023) could allow for continuous updates during the pruning process, preserving model performance. MoE models also provide an interesting direction: while they enable sparse activation, all experts must remain in memory. `Bonsai` could prune individual experts to reduce memory footprint while preserving conditional computation benefits.

Overall, by reducing the memory and computational requirements of structured pruning, `Bonsai` contributes to the growing trend of practical, efficiency-focused AI research, aligning with the community-wide push to make powerful models more accessible. The ability that `Bonsai` enables to prune and adapt large models on a wider range of hardware configurations fosters innovation beyond resource-rich institutions to represent a crucial move toward a more inclusive, sustainable future for the field.

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

## A  Main Experiment Details

### A.1  Hyper-parameters for all `Bonsai` regression during pruning

When using `Bonsai`, we estimate $\beta$ from $\mathbb{D}^l$ by performing linear regression via gradient descent with Adam (Kingma & Ba, 2014). We cross-validate over the following set of hyper-parameters. Note that doing this cross-validation takes much less time than the time needed to construct the dataset $\mathbb{D}^l$. During cross validation,

Table 9: `Bonsai` hyper-parameters for regression. This applies to all experiments unless otherwise specified

| $\gamma$(Regression Weight) | Learning rate | Batch Size | Epochs |
|---|---|---|---|
| {100, 0, 1e-4} | {100, 10, 1, 0.1} | {32, 64, 128} | 50 |

we choose the model whose predictions have the best Kendall rank correlation coefficient (Kendall, 1948) with the target. We do this because we do not care about matching $U_k$ exactly for each sub-model $k$; we rather care that our learned $\beta$ predicts the correct rankings amoung sub-models, which would denote that $\beta$ reasonably models relative module importances.

In general, we use $\ell_1$-norm regularization on $\beta$ for all experiments. For the Phi-2 experiment in Section 4.3, we find that $\ell_2$-norm works better.

### A.2  Forward Pass Only / Semi-structured pruning Experiments

Table 10 shows the `Bonsai` hyperparameters we used for the experiments in Section 4.1.

Table 10: `Bonsai` hyper-params for forward only experiments

Table 11: `Bonsai` fine-tuning HP for pruned LLaMA family models

| $p_{\text{iter}}$ | $\text{ns}_{\text{sub−models}}$ | $\text{ns}_{\text{data}}$ | Metric for $\rho$ | LR | rank | LoRA-$\alpha$ | $\lambda$ (Distill Weight) | LoRA Modules |
|---|---|---|---|---|---|---|---|---|
| 0.05 | 200 | 32 (per-iter) | Wanda | 1e-4 | 128 | 4×rank | 0.01 | All Modules |

For Wanda(Sun et al., 2023), we use the default hyper-parameters specified by the paper repo here for pruning. For fine-tuning, we use rank = 64. We apply LoRA to only the q_proj and v_proj matrices in each layer of the pruned LLaMA model – this is unlike with `Bonsai` where we fine-tune all modules. We cannot do same because since the Wanda model just produces sparse matrices, the matrices instantiated during the backward pass are the same sizes as the sparsified matrices and thus occupy more memory (compared to our approach that actually makes the matrices smaller in dimension instead of sparsifying). We are also unable to perform distillation on the Wanda models due to this reason. For fine-tuning the Phi-2 model on Wikitext-2, we use the same hyper-parameters as `Bonsai` in Table 11.

### A.3  Experiments comparing to Gradient based structured pruning

We compare to LoRA-Prune and LLM-Pruner. We take their performance results directly from the LoRA-Prune paper. Whilst we use 1 A6000 GPU (48G) for all experiments, LoRA-Prune uses A100 GPU (80G) for pruning LLaMA-1 7B.

All `Bonsai` hyper-parameters are the same as Appendix A.2 except for $\text{ns}_{\text{sub−models}}$ which we set to 1000.

### A.4  Phi-2 pruning experiment details

For the experiment in Section 4.3, All `Bonsai` hyper-parameters are the same as Appendix A.2 except for the following changes:

- $\text{ns}_{\text{sub−models}} = 2000$

- $p_{\text{iter}} = 0.35$. We thus perform 1-shot pruning directly to the target sparsity of 35%. We find that this seems to work best for the Phi-2 model. We posit that this might be because the Phi-2 models use LayerNorm (Ba et al., 2016) whilst the other models we explore, LLaMA and Mistral use RMSNorm.

- Due to its relatively small size, the 1.8B pruned model can be fully fine-tuned on a single A6000 GPU over 100k sequences of length 2,048 tokens from the C4 dataset instead of using LoRA.

## B Impact of regression and perturbation ablation details

For the experiment in Appendix I, All `Bonsai` hyper-parameters are the same as Appendix A.2 except $p_{\text{iter}} = 0.1$ to speed up pruning.

A simple alternative to `Bonsai` is to leverage the prior $\rho$, computed from the unperturbed parent model, and make pruning decisions exclusively according to this. This is the `No Perturbation + No Regression` baseline in Figure 3. This approach has quite poor performance. We can further improve this baseline by adding back perturbative aspect where we prune the parent model according to $\rho'$ which is computed by aggregating the $\rho$ metric computed over the **perturbed** models. Note that we use a Wanda based metric to define $\rho$ for this experiment. module-level analogues of the unstructured pruning metrics we explore are defined in Appendix E.

**Table 12:** Experiment on linear regression to estimate module importances. Wikitext-2 Perplexity. LLaMA-2 7B pruned to 50% sparsity

| Linear Regression | Relative Speedup | w/o Post-Pruning Adaptation | w Post-Pruning Adaptation |
|:---:|:---:|:---:|:---:|
| No | 2.06 | 146.57 | 9.68 |
| Yes | 1.77 | 61.63 | 9.15 |

## C Varying the pruning fraction per-iteration

For the experiment in Appendix I, All `Bonsai` hyper-parameters are the same as Appendix A.2 except we vary $p_{\text{iter}}$.

**Table 13:** Varying the fraction pruned at a time. Wikitext-2 Perplexity. LLaMA-2 7B pruned to 50% sparsity

| Prune Frac | Relative Speedup | w/o Post-Pruning Adaptation | w Post-Pruning Adaptation |
|:---:|:---:|:---:|:---:|
| 0.05 | 1.58 | 19.47 | 8.89 |
| 0.1 | 1.77 | 61.63 | 9.15 |
| 0.20 | 1.67 | 209.44 | 9.57 |

## D Varying the number of calibration data points for pruning & PPA

For these two categories of experiments, all `Bonsai` hyper-parameters are the same as Appendix A.2 except we vary $\text{ns}_{\text{data}}$ and $p_{\text{iter}} = 0.1$ to speed up pruning in the former.

## E Impact of prior

For this experiment, All `Bonsai` hyper-parameters are the same as Appendix A.2 except we vary $\rho$.

**Table 14:** How many data points to consider during forward passes. Wikitext-2 Perplexity. Llama-2 7B pruned to 50% sparsity

| $\text{ns}_{\text{data}}$ | w/o Adapt | w Adapt |
|:---:|:---:|:---:|
| 8 | 130.04 | 9.45 |
| 32 | 61.63 | 9.15 |

### E.1   $\rho$ is Activation Magnitude

**MLP / Fully Connected Module**: Let $d$ be the intermediate dimension of the MLP to be pruned. Note that for all transformer models evaluate, the MLP components are 2 layer and thus have a single intermediate dimension. For any data-sample sequence $b$, we flatten model activation at this point $\mathbf{a} \in \mathbb{R}^{B \times S \times d} \to \mathbb{R}^{BS \times d}$ and then compute the following averaged activation magnitude :

$$\left(\rho \in \mathbb{R}^d\right) \propto \hat{\mathbf{a}} = \frac{1}{B} \sum_b \text{Mean}\left(\left|\mathbf{a}_b\right|, \text{axis} =0\right) \tag{4}$$

**Self-Attention Module**: For any data-sample sequence $b$, the output of the self-attention module before the final output projection is $\mathbf{a} \in \mathbb{R}^{B \times S \times d_h \times h}$ where $h$ is the number of attention heads and $d_h$ is the size of each head's output. We can flatten $\mathbf{a} \in \mathbb{R}^{B \times S \times d_h \times h} \to \mathbb{R}^{BSd_h \times h}$ and then use the same formula as Equation 5 above to calculate $\rho$.

$$\left(\rho \in \mathbb{R}^h\right) \propto \hat{\mathbf{a}} \tag{5}$$

### E.2   $\rho$ is Wanda (Sun et al., 2023)

**MLP / Fully Connected Module**: Let $d$ be the intermediate dimension of the MLP to be pruned. Let $W \in \mathbb{R}^{d \times o}$ be the output projection matrix for the MLP. For any data-sample sequence $b$, we flatten model activation before the final output, $\mathbf{a} \in \mathbb{R}^{B \times S \times d} \to \mathbb{R}^{BS \times d}$ and then compute the following metric which is a module-level analogue of Wanda:

$$\left(\rho \in \mathbb{R}^d\right) \propto \hat{\mathbf{a}} = \frac{1}{o} \sum_o \mathbf{a}^o$$
$$\mathbf{a}^o = \left|W[:,o]\right| \odot \text{RootMeanSquare}\left(\mathbf{a}, \text{axis} =0\right) \tag{6}$$

**Self-Attention Module**: Let $W \in \mathbb{R}^{d \times o}$ be the output projection matrix for the self-attention module. For any data-sample sequence $b$, the output of the self-attention module before the final output projection is $\mathbf{a} \in \mathbb{R}^{B \times S \times d_h \times h}$ where $h$ is the number of attention heads and $d_h$ is the size of each head's output. We can flatten $\mathbf{a} \in \mathbb{R}^{B \times S \times d_h \times h} \to \mathbb{R}^{BSd_h \times h}$ and then use the same formula as Equation 5 above to calculate $\rho \in \mathbb{R}^h$.

### E.3   $\rho$ is Fluctuation-based (FLAP) (An et al., 2024)

**MLP / Fully Connected Module**: Let $d$ be the intermediate dimension of the MLP to be pruned. Let $W \in \mathbb{R}^{d \times o}$ be the output projection matrix for the MLP. For any data-sample sequence $b$, we compute the sample variance of each input feature across batches and weight it with the squared norm of the corresponding column of the weight matrix:

$$\left(\rho \in \mathbb{R}^d\right) \propto S_{:,j} = \frac{1}{N-1} \sum_{n=1}^{N} (X_{n,j,:} - \bar{X}_{:,j,:})^2 \cdot \|W_{:,j}\|_2^2 \tag{7}$$

where $\bar{X}_{:,j,:}$ represents the average of the $j$-th channel for all samples, and $\|W_{:,j}\|_2^2$ denotes the squared norm of the $j$-th column of the weight matrix.

**Self-Attention Module**: We compute the fluctuation at the head level, weighted by the corresponding weights in the output projection matrix:

$$\left(\rho \in \mathbb{R}^h\right) \propto S_{:,j} = \frac{1}{N-1} \sum_{n=1}^{N} (X_{n,j,:} - \bar{X}_{:,j,:})^2 \cdot ||W_{:,j}||_2^2 \tag{8}$$

For our improved fluctuation metric (fluct.2.0), we track the mean and second moment separately over multiple batches, computing variance as:

$$\mathrm{Var}(X) = \mathbb{E}[X^2] - \mathbb{E}[X]^2 \tag{9}$$

This approach provides greater numerical stability when accumulating statistics across multiple forward passes.

## F  How many perturbative samples are reasonable?

For this experiment, All `Bonsai` hyper-parameters are the same as Appendix A.2 except $p_{\text{iter}} = 0.1$ to speed up pruning.

**Table 15:** Varying the number of sub-models generated. Wikitext-2 Perplexity. LLaMA-2 7B pruned to 50% sparsity

| Num Samples | w/o Post-Pruning Adaptation | w Post-Pruning Adaptation |
|:---:|:---:|:---:|
| 1000 | 22.09 | 9.25 |
| 200 | 61.63 | 9.15 |
| 50 | NaN | 9.24 |

Using $\text{ns}_{\text{sub-models}} = 50$ results in a model with NaN perplexity on the Wikitext validation set. We posit that this is because of the LLaMA models are in half precision, and removing the wrong modules can result in activations going outside of the FP16 dynamic range for unique data points. Note that we are able to recover good performance of the model after fine-tuning though (we do not observe NaNs with the Wikitext-2 training data). This indicates that `Bonsai` actually recovers good modules even using as few samples as 50 sub-models.

## G  LlaMA-3-8B Experiments

In our efforts toward generalization, we also validate Bonsai on LLaMA-3-8B and use the same experimental setup as our LLaMA-2-7B experiments (hyperparameters in Table A.2) and prune to 50% sparsity on Wikitext-2.

**Table 16:** LLaMA-3-8B at 50% sparsity on Wikitext-2

| Method | Wikitext-2 PPL |
|:---|:---:|
| LLaMA-3-8B (base) | 6.14 |
| Bonsai (w/o PPA) | 21.36 |
| Bonsai (w/ PPA) | 10.37 |

The results show comparable trends to LLaMA-2-7B in the main text: Bonsai operates within accessible memory budgets while achieving competitive accuracy after post-pruning adaptation. This confirms that Bonsai's approach generalizes across LLaMA model versions.

# H    Mistral-7B Experiment Details

In addition to the primary experiments on the LLaMA and Phi-2 models, supplementary experiments were performed on the Mistral-7B Jiang et al. (2023) model in comparison with Wanda results on the stated model. We apply `Bonsai` with the same hardware and configuration settings as used for the LLaMA and Phi-2 experiments. We target different pruning fractions (0.05, 0.1, and 0.2) across different numbers of samples and masks per iteration to evaluate the method's performance under varying sparsity conditions.

The Mistral-7B model architecture differs from the LLaMA architecture in its use of group query attention and sliding window attention in lieu of the standard self-attention used in most transformer-based models like LLaMA Jiang et al. (2023). We factor these differences into consideration in the implementation of `Bonsai` for Mistral. For the experiments that produced the results below, all `Bonsai` hyper-parameters are the same as Appendix A.2.

Table 17 presents the test perplexity results for Mistral-7B under different pruning methods. Considering the fully-structured sparsity nature of `Bonsai`, it achieves a test perplexity of 47.5 without post-pruning adaptation, with 1.66× inference speedup. After performing post-pruning adaptation on our pruned Mistral-7B, perplexity dropped drastically to 10.08. Note that the reported results of Wanda-pruned Mistral-7B are not fine-tuned afterward; if they were, their results would be marginally better than `Bonsai`'s results. However, as shown in Table 3, latency speedup would have dropped rapidly, while `Bonsai` stays the same at 1.66×.

**Table 17:** Test perplexity of Mistral-7B model on Wikitext-2 across fully-structured `Bonsai` and semi-structured Wanda methods.

|  | Sparsity Level | Method | Test PPL |
|---|---|---|---|
| Original, unpruned Mistral-7B | N/A | N/A | 5.245 |
| Wanda | semi-structured 2-4 | magnitude
Wanda
SparseGPT | 13.81
12.38
10.46 |
| `Bonsai` (w/o Adaptation) | structured 50% | magnitude
Wanda | 67.48
47.50 |
| `Bonsai` (w/ Adaptation) | structured 50% | Wanda | 10.08 |

We further investigate the pruning habits of `Bonsai` by examining the pruned layers of Mistral, as shown in Figure 5. We notice a recurring theme: when an attention layer is significantly altered, it leads to compensation in the next layers within the sequence. This adaptive behavior, termed the "Hydra effect" by (McGrath et al., 2023), implies that the layers within a language model interact in a way that changes in one layer prompt adjustments in another. (McGrath et al., 2023) specifically mentioned that when one attention layer was removed from a language model, the model was still able to self-repair and produce similar outputs; but it did so by relying more heavily on other layers.

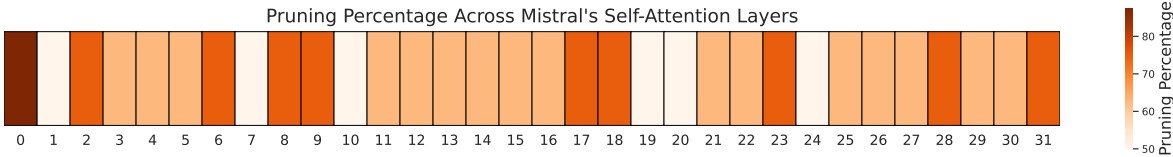

**Figure 5:** Mistral's pruned attention layers. The heavily pruned layers are usually preceded by or sandwiched between lightly-pruned layers, exhibiting the self-repairing "Hydra effect" McGrath et al. (2023).

## I  Should `Bonsai` prune iteratively?

Table 18 demonstrates the benefits of using `Bonsai` in an iterative fashion. Pruning slowly ($p_{\text{iter}} = 0.05$) yields the best results, but this comes at the cost of increasing the total time to prune the model. The performance gap between values of $p_{\text{iter}}$ persists even after post-pruning adaptation, indicating that slower pruning allows for more accurate estimates of module importance.

**Table 18:** Varying $p_{\text{iter}}$. Wikitext-2 perplexity of LLaMA-2 7B pruned to 50% sparsity. See Appendix C for experiment details.

|  | $p_{\text{iter}} = 0.05$ | $p_{\text{iter}} = 0.1$ | $p_{\text{iter}} = 0.2$ |
|---|---|---|---|
| **w/o Adapt** | 19.47 | 61.63 | 209.44 |
| **w Adapt** | 8.89 | 9.15 | 9.57 |

