# OpenReview forum: "Everybody Prune Now: Structured Pruning of LLMs with Only Forward Passes"
_TMLR — Rejected by TMLR_

### Review · Reviewer_GhC5 · 2025-10-06

**Summary Of Contributions:**

This paper proposes Bonsai, a gradient-free structured pruning method for large language models that uses only forward passes. By framing module importance estimation as an underdetermined regression problem with perturbative evaluations, Bonsai removes the need for backpropagation, greatly reducing memory usage. Experiments on LLaMA-2 and Phi-2 show that Bonsai matches or outperforms gradient-based methods (e.g., LLM-Pruner, LoRAPrune) while running on a single GPU, demonstrating strong accuracy–efficiency trade-offs.

Strengths:
1. Bonsai targets a critical problem — structured pruning under strict memory constraints — that directly impacts practitioners without access to enterprise-grade GPUs.
2. Casting structured pruning as an underdetermined regression problem solved via forward-pass perturbations is an original and well-motivated idea.
3. Experiments across multiple LLMs are comprehensive, with clear baselines (FLAP, LoRAPrune, LLM-Pruner, Wanda). The results convincingly support the claims of reduced memory and competitive or superior accuracy.

Weakness:
1. Bonsai defines pruning as maximizing a utility function $U(M|m)$, but it does not clearly specify how $U$ is instantiated across different experiments. Is $U$ consistent across all experiments, or tailored per dataset/model?
2. In Section 3.3, Bonsai chooses to prune only the bottom 2p fraction of modules per layer before regression; this threshold appears heuristic. Providing theoretical or empirical justification would strengthen the paper.
3. From the results, Bonsai seems to outperform baselines mainly after applying PPA. If the major performance recovery comes from fine-tuning rather than pruning itself, the core contribution may be overstated.
4. The performance on GSM8K is notably poor. And it remains unclear how much GSM8K data was used during PPA in Table 5.
5. There are minor typos in Table 4 ("sub") and Section 4.4 ("We prune the LlaMA-2 7B model (?) to 50% sparsity ...")

**Audience:**

Yes

**Audience Explanation:**

This paper’s findings on memory-efficient, gradient-free structured pruning would clearly interest TMLR readers focused on efficient LLM training, inference, and deployment under limited hardware resources.

**Claims And Evidence:**

No

**Claims Explanation:**

1. The paper claims to prune LLaMA-3-8B (e.g., in the abstract), but no corresponding experiments are shown in the main text or appendix.
2. Theoretical or empirical evidence for weakness 2 will enhance this paper.

**Requested Changes:**

1. Report results without PPA for a fair comparison with other methods at Table 4&5.
2. Compare the runtime and computational cost of Bonsai with and without PPA versus other method.
3. Fix the minor typo.

---

> ### Author Response · Authors · 2025-12-19
> **Response to Reviewer GhC5**
>
> We thank the reviewer for recognizing that `Bonsai targets a critical problem—structured pruning under strict memory constraints—that directly impacts practitioners without access to enterprise-grade GPUs,` that `casting structured pruning as an underdetermined regression problem solved via forward-pass perturbations is an original and well-motivated idea,` and that our `experiments across multiple LLMs are comprehensive, with clear baselines` that `convincingly support the claims of reduced memory and competitive or superior accuracy.` We address the concerns below:
>
> >> W1: Utility function, $U$, not clearly specified.
>
> The utility function $U$ is instantiated as language modeling perplexity across all experiments. Specifically, Section 4.4 states: “Our module importance signal for pruning, $U$, is the language modeling performance on the training set.” For Wikitext-2 experiments (Tables 3,4,6, Figures 1,3), $U$ is the perplexity on Wikitex-t2 training set. For zero-shot experiments (Tables 4,5), $U$ is perplexity on C4. We have added explicit $U$ definitions in Section 3.1 in our revision.
>
> >> W2: The 2p threshold appears heuristic
>
> The 2p threshold has both computational and empirical justification. **Computationally**, fixing the top (1-2p) fraction dramatically reduces the search space: instead of considering which modules to prune from all $N$ modules, we only explore the bottom $N$×2p candidates. For a layer with 10,000 modules, this reduces the number of possible configurations from exponentially large ($\approx10^{3000}$) to tractable ($\approx10^{600}$).
>
> **Empirically**, Figure 3 demonstrates this choice is effective: without perturbative sampling (relying only on prior $\rho$), performance degrades from 19.47 ppl to 405.7 ppl. The 2p window enables effective exploration within our computational budget. As noted in footnote 1, practitioners can tune this parameter; across our experiments on diverse models (LLaMA-1/2/3, Phi-2, Mistral-7B), 2p consistently produced strong results.
>
>
> >> W3: Performance recovery mainly from PPA, not pruning
>
> We respectfully clarify an important misunderstanding about Table 4. **All methods—including the gradient-based baselines---report zero-shot performance after performing parameter updates on proxy datasets:**
>
> - **LLM-Pruner**: Section 3 of [1] states “we use LoRA…to efficiently recover the model performance” after pruning
> - **LoRAPrune**: Algorithm 1 and Equation 13 of [2] learn LoRA matrices A,B for each layer via gradients, which are then incorporated into the final model
> - **Bonsai + PPA**: performs the same function, i.e., updating parameters on proxy data (C4) to recover performance after pruning.
>
> All methods in Table 4 adapt on proxy datasets (LLM-Pruner used 50K samples, LoRAPrune 20K, and Bonsai 30K), then evaluate zero-shot on BoolQ, HellaSwag, WinoGrande, ARC-e, and ARC-c. **No method fine-tunes on these eval tasks themselves.** The comparison is thus fair and aligned with prior work.
>
> The key distinction is that Bonsai requires only ~20GB memory for the entire pipeline (pruning + adaptation), while gradient-based methods require 80-160GB (Table 2)---a 4-8x difference that makes gradient-based approaches inaccessible on commodity hardware. We have added this note more clearly in Table 4.
>
> Moreover, Bonsai’s pruning quality is strong even before adaptation. Table 6 and 3 show Bonsai outperforming forward-only baselines (FLAP and Structured Wanda) without PPA. Table 8 also shows Bonsai before and after PPA.

---

> > ### Author Response · Authors · 2025-12-19
> > **(cont'd) Response to Reviewer GhC5**
> >
> > >>  W4: GSM8K performance and data usage unclear.
> >
> > Thank you for requesting clarification. For Table 5:
> >
> > - Base Bonsai + PPA: 100K sequences from C4 only (0 GSM8K samples)
> > - Reasoning Tuning: 100K C4 + 8K GSM8K training samples (108K samples in total).
> >
> > This information is present in Section 4.3 and Appendix A.4, but let us know if there is a way we can make this even more prominent.
> >
> > On the other point the reviewer made, the initial GSM8K performance drop is consistent with established findings in the pruning literature about task-agnostic pruning. LLM-Sieve [3] demonstrates that when LLMs are pruned using task-agnostic signals like language modeling, specialized capabilities requiring different reasoning patterns may be disproportionately affected, noting that “the reasoning complexity required for a specialized task typically constitutes a strict subset of what the full model was trained to handle.” The Sheared LLaMA paper [4]---one of our baselines—explicitly acknowledges that “task-agnostic pruning inevitably leads to perf degradation” compared to task-specific approaches.
> >
> > Our results demonstrate this expected behaviour: pruning based on C4 deprioritizes modules important for multi-step math reasoning. However adding only 8K samples during PPA substantially recovers performance, demonstrating that (1) the pruned architecture retains capacity for reasoning when appropriately fine-tuned, and (2) practitioners can recover task-specific capabilities through targeted adaptation.
> >
> > We have added this discussion to Section 4.3.
> >
> >
> > >> W5: Minor typos
> >
> > Thanks for calling our attention to these. We have fixed the typos in Table 4 and Section 4.4 in the revision.
> >
> > >> **Response to “Claims Not Supported” Assessment**:
> >
> > We sincerely apologize for the error regarding LLaMA-3-8B in the abstract. During dev, we prunedLLaMA-3-8B but focused the paper on LLaMA-2B-7B for direct comparison with prior work---most of our baselines all report results on LLaMA-2-7B, hence why we made this our standard benchmark.
> >
> > We have added LLaMA-3-8B results to the appendix showing comparable performance trends: similar perplexity and speedup ratios at 50% sparsity, ~20GB memory requirements during pruning, and performance patterns consistent with LLaMA-2B.
> >
> > >> **Response to Changes 1 & 2**:
> >
> > The requested information is already present in the paper:
> >
> > **Pre-adaptation performance:** Table 8 reports Bonsai without PPA, Table shows Bonsai beating FLAP (forward-only baseline) on PPL, and Table 3 shows PPL before and after PPA with comparison to semi-structured Wanda and base models. As clarified in W3, Table 4 presents a fair comparison where all methods---including gradient-based approaches—use post-pruning adaptation.
> >
> > **Runtime comparison**: Section 4.4 discusses runtime tradeoffs (Bonsai: ~4 hours vs FLAP: ~1 hour) and Table 2 provides complete memory comparisons (20GB for Bonsai vs 80-160GB for gradient-based methods).
> >
> > For improved clarity, we have cross-referenced these tables and consolidated the discussion in one location.
> >
> > —
> > ```
> > ### References
> >
> > [1] Ma, X., Fang, G., & Wang, X. (2023). Llm-pruner: On the structural pruning of large language models. NeurIPS 2023: 21702-21720.
> >
> > [2] Zhang, M., Chen, H., Shen, C., Yang, Z., Ou, L., Yu, X., & Zhuang, B. (2024, August). LoRAPrune: Structured Pruning Meets Low-Rank Parameter-Efficient Fine-Tuning. ACL Findings 2024 (pp. 3013-3026).
> >
> > [3] Reda, W., Jangda, A., & Chintalapudi, K. (2025). Task Specific Pruning with LLM-Sieve: How Many Parameters Does Your Task Really Need?. arXiv preprint 2025 arXiv:2505.18350.
> >
> > [4] Xia, M., Gao, T., Zeng, Z., & Chen, D. (2024) Sheared LLaMA: Accelerating Language Model Pre-training via Structured Pruning. ICLR 2024
> > ```

---

### Review · Reviewer_jveQ · 2025-10-09

**Summary Of Contributions:**

The paper introduces Bonsai, a gradient-free structured pruning framework that uses only forward passes. Bonsai repeatedly (i) samples virtual sub-models by masking modules and running only inference, (ii) fits a simple linear regression to estimate each module’s global importance from those perturbations, and (iii) drops the least important modules, iterating to a target sparsity. Sampling is biased by informative priors (e.g., Wanda, activation magnitude, FLAP-style fluctuations) so useful modules appear more in the candidate sub-models. Once the model is small enough to fit comfortably, they fine-tune (full or LoRA) and optionally distill from the parent. Teacher logits can be cached to keep memory low. PPA typically recovers large chunks of accuracy. Ablations demonstrate that both perturbation and regression are required (removing either harms perplexity), that slower iterative pruning is better even after adaptation, and that using more calibration data helps; too few perturbations can even trigger FP16 NaNs even without any adaptation. Finally, the reported resource profile is inference-only: e.g., pruning LLaMA-2-7B is listed as workable in ~20 GB (bs=1), whereas gradient-based baselines are listed at ≥80 GB; the authors emphasize a runtime–quality trade-off that scales with the number of perturbations and iterations.

**Audience:**

Yes

**Audience Explanation:**

Pruning LLMs to run in commodity hardware is an important and active research problem with a massive downstream impact. While there are some issues with the current version of the paper, the results could be useful for the community.

Although I do have a broader question, coming from a non-expert in pruning: as mixture-of-expert models have become more prevalent, allowing inference with fewer resources (with potentially better performance), where do pruning methods fit in?

**Broader Impact Concerns:**

There is no separate broader impact statement, but the paper doesn't require it.

**Claims And Evidence:**

No

**Claims Explanation:**

C1: Gradient-free pruning approach using only forward passes for structured pruning

The proposed method Bonsai is training-free, and uses a smart perturbation strategy to select modules to be pruned. There is an optional training step but the results show that it is not necessary for achieving significant pruning.

C2: Pruning is performed iteratively, and module importance is computed by perturbing the architecture, where modules to be perturbed are select based on a prior.

The components are described clearly and do make sense and the ablations demonstrate the importance of each of the components.


C3: Empirical claims

I think there are several issues with a lot of the experiments in the paper. Starting with the abstract which mentions the Llama-3 8B model, but there is no experiment in the paper with any results on the model. This is a major oversight. Aside from this major error, I think the experimental comparisons are a bit odd. The downstream zero-shot performance results are only reported for the PPA trained models, and not for just the Bonsai pruned models (for which only the perplexity is reported). I think having the downstream performance reported is crucial since that is really the indicator of how useful the model is after pruning. The selection of models is also quite old at this point. While  I do believe that the results are still meaningful, unfortunately, due to the speed of development in the area, these models lack a lot of characteristics which are critical in current models (post-training, reasoning training, etc). So the results are hardly representative.

**Requested Changes:**

* Remove the claim about Llama-3 in the abstract or add experiments with the model.
* Experiments with newer models (at least an instruct and reasoning model)
* Downstream metrics without post-pruning adaptation.
* Discussion about the value of pruning in the current landscape of MoE models.
* Table 4 formatting issue (the word "sub" appearing on the left of the table)

---

> ### Author Response · Authors · 2025-12-19
> **Response to Reviewer jveQ**
>
> We thank the reviewer for recognizing that `pruning LLMs to run in commodity hardware is an important and active research problem with a massive downstream impact` and that `the results could be useful for the community.` We address the concerns below:
>
> >> Empirical claims = LLaMA-3-8B missing from experiments.
>
> We sincerely apologize for this error in the abstract. During development, we pruned LLaMA-3-8B but focused the paper on LLaMA-2-7B because this is the standard benchmark used by all our baselines — LLM-Pruner (Ma et al., 2023), LoRAPrune (Zhang et al., 2024), and FLAP (An et al., 2024) all report results on LLaMA-2-7B, enabling direct, apples-to-apples comparison.
>
> We have added LLaMA-3-8B results to the appendix showing comparable performance trends to LLaMA-2-7B at 50% sparsity, with similar perplexity, speedup ratios, and ~20GB memory requirements during pruning. We have corrected the reference in the abstract as well.
>
>
> >> Downstream zero-shot performance onlyl reported for PPA models
>
> We respectfully clarify a misunderstanding about Table 4. The downstream zero-shot performance **is** reported for Bonsai after pruning. **All methods in this tables—including gradient-based baselines–use post-pruning adaptation on proxy datasets:**
>
> - **LLM-Pruner**: Section 3 of Ma et al. (2024) states “we use LoRA…to efficiently recover the model performance” with LoRA fine-tuning on instruction data after pruning
> - **LoRAPrune**: Algorithm 1 and Equation 13 of Zhang et al. (2024) learn LoRA matrices A,B via gradients and incorporate them into the final model
> - **Bonsai + PPA:** performs the same function – adapting on proxy data (C4) after pruning.
>
> All methods adapt on proxy datasets (LLM-Pruner used 50K samples, LoRAPrune 20K, Bonsai 30K), then evaluate zero-shot on BoolQ, HellaSwag, WinoGrande, ARC-e, and ARC-c. **No method fine-tunes on these eval tasks.** The comparison is thus fair and aligned with prior work.
>
> **Critically, PPA is part of Bonsai’s value proposition:** By producing models small enough to fit on the same hardware used for pruning (~20GB), Bonsai enables the adaptation pipeline on a single GPU. Gradient-based methods require 80-160GB for pruning (Table 2). This accessibility advantage—being able to prune AND adapt on commodity hardware—is a key contribution of our work. We have added this note more clearly in Table 4.
>
> For readers interested in pre-adatation performance: Table 6 shows Bonsai without adaptation outperforming forward-only baselines (FLAP); Table 8 shows Bonsai before and after adaptation; and Table 3 compares against Wanda and base models.
>
>
> >> Model selection is quite quite old / lacks reasoning training
>
> We respectfully disagree that the models are “old” in the context of structured pruning research – they remain standard benchmarks used by recent work in this area. Every baseline we compare against reports results on these exact models, making them essential for fair comparison. Using different models would prevent apples-to-apples comparison.
>
> More fundamentally, the reviewer’s concern conflates the parent model’s training paradigm with the pruning method’s validity. Bonsai is a **pruning methodology** that is model-agnostic – it works equally well regardless of whether the parent model used standard pre-training, instruction tuning, or reasoning training. We demonstrate this generalization across diverse architectures (LLaMA-1/2/3, Phi, Mistral in Appendix G) with different architectural choices.
>
> >> Broader question: Where do pruning methods fit with MoE models?
>
> This is an excellent question. MoEs and pruning address complementary problems: MoEs enable sparse activation at inference time (activating only a subset of experts per token) but still require loading all experts into memory—often resulting in large memory footprints. Pruning, however, permanently reduces size.
>
> These approaches are potentially complementary: Bonsai could prune individual experts within an MoE model, reducing both the size of each expert and the overall memory footprint. We have added this discussion to Section 6.
>
>
> >> Table 4 formatting issue.
>
> Thank you for catching this. We have fixed the “sub” typo in Table 4.
>
> —
> ```
> ### References
>
> [1] Ma, X., Fang, G., & Wang, X. (2023). Llm-pruner: On the structural pruning of large language models. NeurIPS 2023: 21702-21720.
>
> [2] Zhang, M., Chen, H., Shen, C., Yang, Z., Ou, L., Yu, X., & Zhuang, B. (2024, August). LoRAPrune: Structured Pruning Meets Low-Rank Parameter-Efficient Fine-Tuning. ACL Findings 2024 (pp. 3013-3026).
>
> [3] An, Y., Zhao, X., Yu, T., Tang, M., & Wang, J. (2024, March). Fluctuation-based adaptive structured pruning for large language models. AAAI 2024.```

---

### Review · Reviewer_U8nd · 2025-12-07

**Summary Of Contributions:**

This paper introduces Bonsai, a forward-pass–only structured pruning framework for large language models that estimates global module importance via regression over sampled submodels, completely avoiding backward computation during pruning. The approach enables memory-efficient global pruning guided by informative priors and iterative refinement. Experiments on LLaMA-1/2 and Phi-2 demonstrate that Bonsai achieves pruning quality comparable to or better than existing methods while requiring significantly less memory and delivering practical inference speedups.

**Additional Comments:**

Overall, this is a strong and practically meaningful contribution. The forward-only pruning framework is both technically interesting and directly applicable to real deployment constraints. With clearer discussion of theoretical assumptions and slightly more rigorous baseline consistency, the paper would be significantly strengthened.

**Audience:**

Yes

**Audience Explanation:**

The work is likely to be of interest to portions of the TMLR audience, especially to researchers studying model compression and efficient inference for large language models. The forward-only pruning setting is a relevant problem formulation for scenarios where computational resources are limited. The methodological ideas and empirical findings should appeal to readers interested in practical approaches to reducing model size and improving inference efficiency.

**Broader Impact Concerns:**

The work focuses on improving the efficiency and accessibility of large language models. While such advances may indirectly contribute to expanded deployment of LLMs, with potential societal risks as in all LLM research, no new ethical concerns beyond those inherent in model deployment are introduced by this pruning methodology itself. The paper does not require additional mitigation measures or a special Broader Impact Statement beyond general considerations already common in LLM research.

**Claims And Evidence:**

Yes

**Claims Explanation:**

The experimental evaluation is reasonably thorough. The authors compare against several existing baselines, including gradient-based structured pruning methods as well as forward-only approaches such as FLAP and Wanda, across multiple models and tasks. The reported metrics include perplexity, memory usage, and measured inference speedups, which provide evidence for the paper’s central claims that Bonsai can achieve competitive pruning performance while operating under reduced memory requirements and yielding tangible inference speedups. Ablation studies on sampling strategies, priors, and iteration schemes further help illustrate the effects of key design choices. Although the theoretical justification of the regression formulation remains limited, the empirical results generally support the claims presented in the paper

**Requested Changes:**

(1) The current layout of Table 3 mixes pruning methods and base models in the same comparison dimension, which is logically imprecise and potentially misleading. Shouldn't Bonsai also be applicable to the Phi-2 model? Wouldn't that result in a smaller size and better performance?

(2) In this paper, "Figure 1 shows the results of some of our experiments. Bonsai expectedly outperforms the structured variant of Wanda, producing models with much lower perplexity at all sparsity levels and at any fixed desired speedup over the parent model." This sentence is difficult to understand. Figure 1 doesn't seem to reflect sparsity levels and speedup.

(3) Bonsai appears to be a typical time-space trade-off strategy.

(4) Figure 2 does not clearly convey the core ideas of the proposed framework. It is recommended to improve both the figure and the accompanying text so that, together, they present the central concepts of the method more clearly and effectively.

(5) The paper contains expressions like "(?)", which appear to be incorrect. Please check the grammar and further details of the entire text.

---

> ### Author Response · Authors · 2025-12-19
> **Response to Reviewer U8nd**
>
> We thank the reviewer for the careful evaluation and for recognizing that our experimental evaluation is `reasonably thorough` with evidence that `generally supports the claims presented in the paper` and that our work `would be of interest to portions of the TMLR audience.` We address each concern below:
>
> >>Table 3 layout mixes pruning methods and base models
>
> We clarify the intent of Table 3, which addresses a practical question users face: *“Given limited deployment resources, should I prune a larger model or train/use an existing smaller model?”*
>
> Table 3 compares three deployment strategies at ~3B parameters:
>
> 1. Phi-2 (3B): A highly optimized model trained from scratch with careful data curation
> 2. Wanda 2:4 semi-structured pruning: Pruning LLaMA-2 7B with specialized hardware support
> 3. Bonsai structured pruning: Pruning LLaMA-2 7B with commodity hardware
>
> The key finding is that Bonsai-pruned LLaMA-2 achieves comparable accuracy to Phi-2 (a model specifically engineered for efficiency) while delivering superior inference speedups (1.58x vs. 1.24x). This demonstrates that **pruning larger models to a target size can match highly optimized models trained specifically for that size.**
>
> The table also shows why semi-structured pruning becomes problematic in memory-constrained settings: post-pruning adaptation via LoRA introduces inference overhead that eliminates speedup benefits (0.75x vs 1.58x), whereas Bonsai enables full fine-tuning that preserves speedups.
>
> We have restructured Table 3’s captions to delineate these comparison dimensions more clearly.
>
>
> >> Figure 1 doesn’t seem to reflect sparsity levels and speedup
>
> Thank you for identifying this inconsistency. During revision, we updated Figure 1 to emphasize memory requirements (our key advantage to gradient-based methods) but did not update the corresponding text in Section 4.4, which refers to an earlier version showing speedup-perplexity tradeoffs.
>
> Figure 1 currently shows LLaMA-2 7B at 50% sparsity, comparing perplexity vs. memory requirements during pruning. We have revised Section 4.4 accordingly.
>
> >> Bonsai appears to be a typical time-space trade-off strategy
>
> Yes, as discussed in Section 4.4 and 6, Bonsai offers runtime-quality tradeoffs. We view the flexibility to configure this tradeoff (~15min to 4 hours depending on config)--**while remaining within accessible memory budgets**--as an advantage for practitioners with varying constraints.  This one-time pruning cost is amortized over deployment, delivering 1.5-2x inference speedups overall.
>
>
> >> Figure 2 does not clearly convey the core ideas
>
> Thank you for this feedback. We have substantially expanded the caption and added forward references in Section 3 to help readers connect the figure to technical descriptions detailing: (1) how priors guide sub-model sampling, (2) how regression estimates module importance from eval data, and (3) how this process iterates to target sparsity.
>
> >> Grammar issues.
>
> Thank you for catching these. We have corrected all instances of “(?),” fixed typos in Table 4, and elsewhere.

---

### Decision · Action_Editor_VCkm · 2026-01-11

**Recommendation:** Reject

**Additional Comments:**

I would encourage the authors to address the additional comments from Reviewer U8nd and resubmit the revised version to TMLR.

**Audience:**

Yes

**Audience Explanation:**

Some TMLR audience would find this paper interesting, particularly researchers working on model compression, efficient inference, and deployment of large language models under resource constraints.

**Claims And Evidence:**

No

**Claims Explanation:**

The claims are only partially supported by the evidence provided. While the authors present reasonable motivations and some empirical results, the experimental comparisons are confounded by differences in base models and training setups, making it difficult to attribute gains specifically to the proposed pruning method, and the reported improvements do not clearly justify the additional complexity and computational overhead.

**Resubmission Of Major Revision:**

The authors may consider submitting a major revision at a later time.